# Reversible and selective ion intercalation through the top surface of few-layer MoS$_2$

Jinsong Zhang[1,2], Ankun Yang [1], Xi Wu[3], Jorik van de Groep[1], Peizhe Tang[4], Shaorui Li[2], Bofei Liu[1], Feifei Shi[1], Jiayu Wan[1], Qitong Li[1], Yongming Sun[1], Zhiyi Lu[1], Xueli Zheng [1], Guangmin Zhou[1], Chun-Lan Wu[1], Shou-Cheng Zhang[4,5], Mark L. Brongersma[1], Jia Li [3] & Yi Cui [1,5]

Electrochemical intercalation of ions into the van der Waals gap of two-dimensional (2D) layered materials is a promising low-temperature synthesis strategy to tune their physical and chemical properties. It is widely believed that ions prefer intercalation into the van der Waals gap through the edges of the 2D flake, which generally causes wrinkling and distortion. Here we demonstrate that the ions can also intercalate through the top surface of few-layer MoS$_2$ and this type of intercalation is more reversible and stable compared to the intercalation through the edges. Density functional theory calculations show that this intercalation is enabled by the existence of natural defects in exfoliated MoS$_2$ flakes. Furthermore, we reveal that sealed-edge MoS$_2$ allows intercalation of small alkali metal ions (*e.g.*, Li$^+$ and Na$^+$) and rejects large ions (e.g., K$^+$). These findings imply potential applications in developing functional 2D-material-based devices with high tunability and ion selectivity.

[1] Department of Materials Science and Engineering, Stanford University, Stanford, California 94305, USA. [2] State Key Laboratory of Low Dimensional Quantum Physics, Department of Physics, Tsinghua University, Beijing 100084, P.R. China. [3] Laboratory for Computational Materials Engineering, Division of Energy and Environment, Graduate School at Shenzhen, Tsinghua University, Shenzhen 518055, P.R. China. [4] Department of Physics, Stanford University, Stanford, California 94305, USA. [5] Stanford Institute for Materials and Energy Sciences, SLAC National Accelerator Laboratory, 2575 Sand Hill Road, Menlo Park, California 94025, USA. These authors contributed equally: Jinsong Zhang, Ankun Yang, Xi Wu. Correspondence and requests for materials should be addressed to Y.C. (email: yicui@stanford.edu)

Two-dimensional (2D) materials such as graphene, hexagonal boron nitride, and transition metal dichalcogenides (TMDs) have attracted intense interest in areas of optoelectronics[1,2], nanoelectronics[3], and membrane separations[4,5], due to their unique physical and chemical properties[6,7]. Molybdenum disulfide (MoS$_2$) is a member of 2D layered TMDs consisting of molecular layers held together by van der Waals forces. Monolayer MoS$_2$ is one of the thinnest semiconductors available and has been widely studied in electronic devices[2,3]. The maximum carrier density is on the order of $10^{12}$–$10^{14}$ cm$^{-2}$ with solid dielectric gating and ionic liquid gating[8,9]. In contrast, few-layer or bulk MoS$_2$ has been mostly used in energy storage[10] and electrocatalysis[11]. Recently, based on the same principle (intercalation/de-intercalation) as in electrochemical applications, guest species such as alkali metal ions (Li$^+$, Na$^+$, and K$^+$) have been introduced into the large interlayer spacing (~ 0.615 nm) to manipulate and optimize the optical and electrical properties of few-layer MoS$_2$[12–14]. Ion intercalation[12,15–24] enables extremely high doping level (e.g., $6 \times 10^{14}$ cm$^{-2}$ in few layer graphene after Li intercalation[25]) compared with electrical gating. Such high doping levels allow new physics to be discovered, such as superconductivity[26] and facilitates applications of few-layer MoS$_2$ in optoelectronic and nanoelectronic devices. However, intercalation of ions often induces wrinkling and distortion of MoS$_2$ and even irreversible structural changes[14,27,28] that hinder its practical applications.

On the other hand, ultrathin 2D materials have been explored as novel separation membrane to realize ultrafast and high-selective sieving of gases and ions at low energy cost[4,5,29–31]. In these applications, the structural defects, interlayer spacing, or pores created by ion bombardment and oxidative etching have been used as the transport channels for the species. Manipulating ion transport through 2D material membrane via delicate electrical control, which has never been achieved before, would be much more effective and provide additional freedom for membrane designs of future functional devices.

Here, we demonstrate reversible and selective ion intercalation through the top surface of few-layer MoS$_2$. We seal the edges of MoS$_2$ to alleviate the structural deformation and to allow careful examination of intercalation only through the top surface. Through in situ optical and Raman measurements as well as the ab-initio density functional theory (DFT) calculations, we prove that the ions can intercalate through the intrinsic defects into the few-layer MoS$_2$, and this type of intercalation is much more reversible than through the edges. The subtle electrochemical control can dramatically modify the optical and electrical properties of MoS$_2$ in a reversible manner. Particularly, we obtain electron density up to $10^{22}$ cm$^{-3}$ in few-nanometer-thick flakes, highest value among all the gating methods. The reversible ion intercalation and de-intercalation through top surface will benefit future material designs in highly tunable and stable 2D material-based optoelectronic and nanoelectronic devices. Furthermore, we show that the sealed MoS$_2$ flakes allow intercalation of Li$^+$, Na$^+$ but not K$^+$ and the selective intercalation through electrochemical control holds great potential in applications, such as ionic sieving and desalination of salted water.

## Results

**Electrochemical intercalation.** Figure 1a shows the layered structure of MoS$_2$. Each layer has a plane of close-packed molybdenum (Mo) atoms sandwiched by two planes of close-packed sulfur (S) atoms (S–Mo–S). The atoms within each layer are strongly bonded by covalent interactions, whereas the interactions between layers are through weak van der Waals forces. The weak forces between the layers allow expansion of the van der Waals gap and insertion of ions. A planar battery configuration (Fig. 1b) was applied to perform electrochemical intercalation where MoS$_2$ flakes and alkali metals or alkali-containing electrode materials were used as working and counter electrodes, respectively. To compare the intercalation behavior of MoS$_2$ flakes with sealed edge and open edge, the Au electrodes around flake or on flake were carefully designed and patterned by electron-beam lithography (EBL) (see Methods). MoS$_2$ flakes, alkali metals/salts, and electrolyte were sealed in a cell where the top transparent glass allows in situ optical observation and confocal Raman microscopy. Figure 1c, d show how sealed edge and open edge configurations are designed on two typical MoS$_2$ flakes

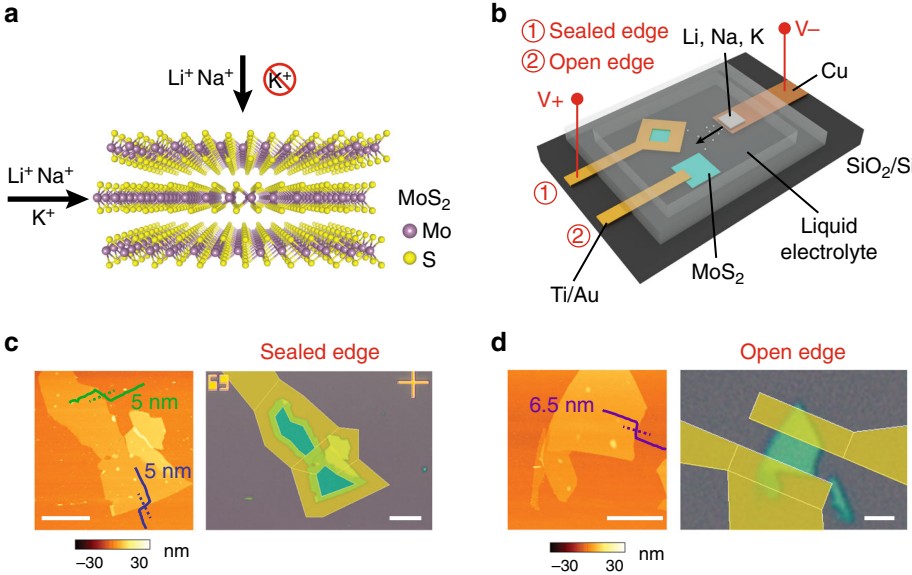

**Fig. 1** Schematic representation of the material and of the experimental setup. **a** Li$^+$, Na$^+$, and K$^+$ intercalation into MoS$_2$ through top and edge channels. **b** An electrochemical cell used to perform intercalation of Li$^+$, Na$^+$, and K$^+$ into MoS$_2$ with sealed and open edge. **c-d** AFM and optical microscopy images of typical MoS$_2$ flakes designed for sealed edge measurement and open edge measurement, respectively. The thicknesses are typically <10 nm, measured across three edges indicated by the dash lines. Scale bars in **c** and **d**, 5 μm

—the electrodes covering all the edges for sealed edge and that only covering part of the flakes for open edge. Atomic force microscopy (AFM) images of the flakes show that these flakes were around 6.5 nm (~10 layers) and 5 nm (~8 layers) in thickness, respectively. We also performed AFM measurements before and after the Au depositions to inspect the $MoS_2$/Au morphology and interface. The clear steps of the Au metal on top of $MoS_2$ edges indicated that the Au deposition was mild and uniform and successfully sealed the $MoS_2$ flake without damaging the covered $MoS_2$ surface (Supplementary Fig. 1).

**Li-ion intercalation into $MoS_2$ through top surface and edges.** Figure 2 shows intercalation of $Li^+$ into $MoS_2$, comparing flakes with sealed edge (i.e., through top surface) and open edge (i.e., through edges). The open circuit voltage (OCV) of $MoS_2$ vs. Li/Li$^+$ was ~3.0 V. We gradually lowered the $MoS_2$ potential with respect to Li/Li$^+$ from 3.5 to 0.8 V with steps of ~0.2 V. We did not go lower than 0.8 V to avoid irreversible conversion reactions[32]. Figure 2a depicts the intercalation of $Li^+$ into the sealed-edge $MoS_2$ flake. When the potential of $MoS_2$ was lowered to around 1.2 V, the color started to change to dark green, indicating the successful intercalation of $Li^+$ into the van der Waals gap of $MoS_2$ from the top surface. The flake exhibited a gradual and uniform change in color with further decrease in potential till 0.8

V, and recovered to its original bright green color when the potential was returned to 3.5 V. The intercalation and de-intercalation can be repeated for several times without disrupting the host structure (Fig. 2a, cycles 2–3). For the open-edge $MoS_2$ flake (Fig. 2b), the color started to change at a relatively high potential around 1.4 V, possibly due to a lower energy barrier for the $Li^+$ to initiate intercalation. The color change started from the edges toward the center, indicative of the preferential intercalation through the edges[14]. With further decrease in the $MoS_2$ potential, in contrast to the sealed-edge $MoS_2$, the color change exhibited non-uniform distribution with two open edges darker than the center of the flake. Furthermore, the color was not recovered when the potential was increased back to 3.5 V. The process was not reversible even when we stopped at a relatively higher potential (e.g., 1.1 V or 1.0 V) (Supplementary Fig. 2)[14]. We attribute the reversible intercalation in sealed-edge $MoS_2$ to three reasons: (1) the flake was clamped and stabilized by the surrounding electrodes preventing structural deformation at the edges of the flake; (2) the diffusion pathways on the top surface are uniformly distributed and mechanically inextensible, which naturally control the intercalation homogeneity compared with the intercalation through the edges where all ions flooded into the opening van der Waals gaps; and (3) the relatively low intercalation rate of sealed-edge $MoS_2$ may cause less lattice distortion and expansion. Although these factors cannot be separated, we

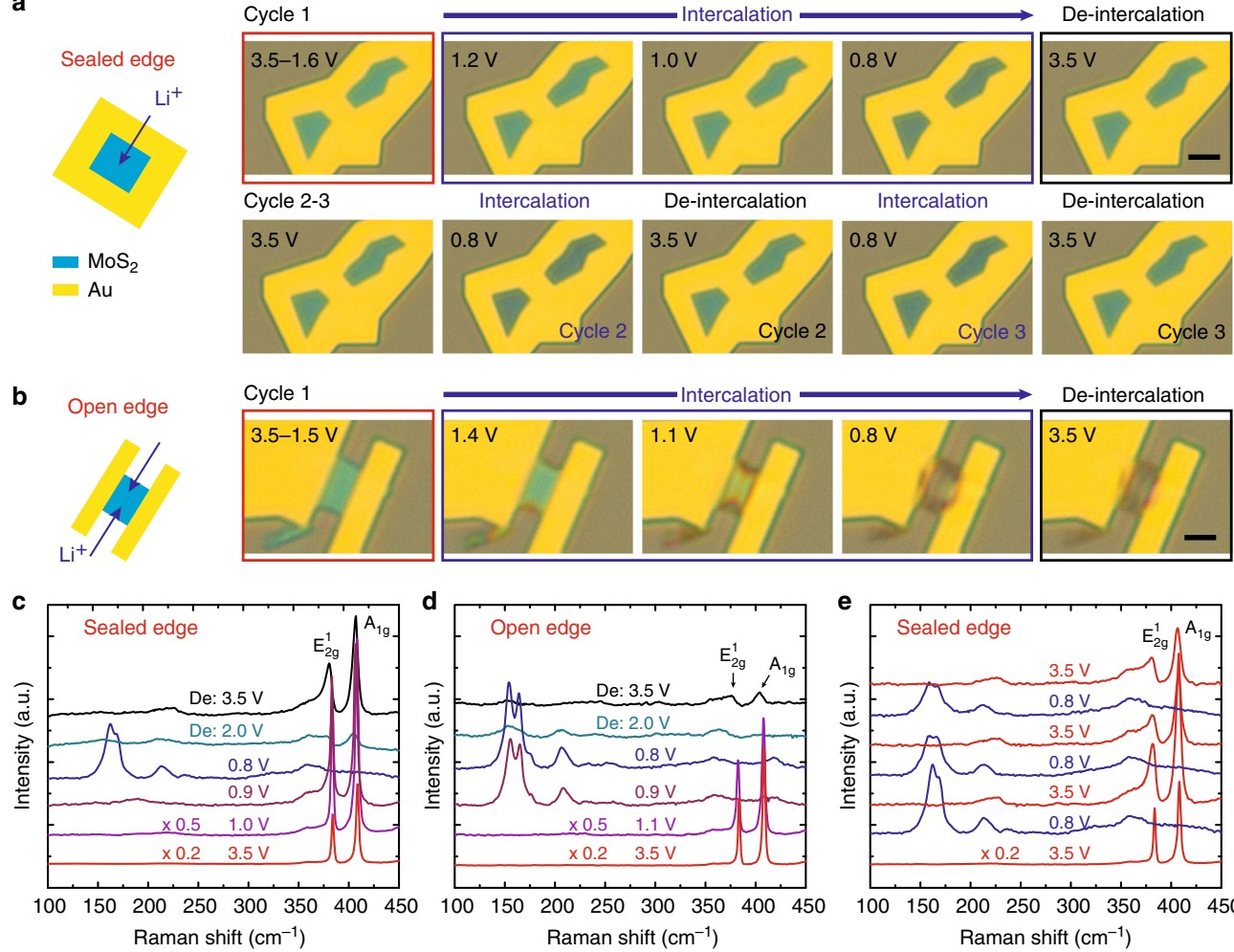

**Fig. 2** Li intercalation of $MoS_2$ through the top surface and edges. **a–b** In situ optical microscopy images of Li intercalation into $MoS_2$ through top surface and edges, respectively. Scale bars, 5 μm. **c–d** In situ Raman spectra of Li intercalation into $MoS_2$ with sealed edge and open edge, respectively. **e** Three cycles of in situ Raman spectra of Li intercalation $MoS_2$ with sealed edge. The voltages were with respect to Li/Li$^+$

believe the mechanical stabilization from surrounding Au electrode contributes more to the reversibility. Because the diffusion pathways on the top surface are naturally present in the $MoS_2$ flakes, and these can only be triggered when all the edges are sealed.

We performed in situ Raman spectroscopy to compare the changes in $MoS_2$ during $Li^+$ intercalation for sealed-edge and open-edge configurations. At 3.5 V, both the Raman spectra showed two peaks located at ~384 and 408 $cm^{-1}$, corresponding to the $E_{2g}^1$ (in-plane optical vibrations of the Mo–S bond) and $A_{1g}$ (out-of-plane optical vibration of S atoms) modes of $MoS_2$ (Fig. 2c, d). Besides the shift of $E_{2g}^1$ and $A_{1g}$ peaks due to gating effect (Supplementary Fig. 3), the Raman spectra did not show obvious change until 1.0 V for sealed-edge $MoS_2$ (Fig. 2c) and 1.1 V for open-edge $MoS_2$ (Fig. 2d). Below these critical potential values, the spectra exhibited significant change and two pronounced differences can be observed by comparing the two sets of Raman spectra. First, the emergence of the new Raman modes at 154, 164, and 207 $cm^{-1}$, often associated with the 2 H to 1 T phase transition[11,14,33], occurs at higher potentials (~0.9 V) for open-edge $MoS_2$ than for sealed-edge $MoS_2$ (~0.8 V). Second, the $E_{2g}^1$ and $A_{1g}$ Raman modes were largely restored and well-defined for the sealed-edge $MoS_2$ after de-intercalation; in contrast, the same two Raman peaks of the open-edge nearly disappear after de-intercalation. These observations were consistent with the differences shown in the optical images and confirmed that sealed-edge $MoS_2$ can show stable and reversible intercalation. By carefully comparing the peak positions of the Raman modes of 1 T phase, we found that the degree of Li intercalation in open-edge $MoS_2$ was slightly higher than that of the sealed-edge $MoS_2$. This is due to the lower energy barrier for intercalation in open-edge $MoS_2$, which makes it hard to control the intercalation process and partially accounts for the irreversibility. Still, even for the sealed-edge $MoS_2$, the modes dropped in intensity (to ~20% of the pristine $MoS_2$) and showed a broadened line-width, which was likely due to the presence of residual strains[34].

To further demonstrate the robustness of the sealed-edge $MoS_2$ geometry, in situ Raman spectra corresponding to the three cycles of intercalation and de-intercalation were recorded (Fig. 2e). These spectra were highly reproducible showing well-defined $E_{2g}^1$ and $A_{1g}$ Raman modes before intercalation and after de-intercalation, as well as the new Raman modes around 170 and 220 $cm^{-1}$ after intercalation. In addition, we found both the $E_{2g}^1$ and $A_{1g}$ Raman modes can be clearly identified for up to the 20 cycles of Li intercalation and de-intercalation processes (Supplementary Fig. 4), indicating the crystalline stability of sealed-edge $MoS_2$. Furthermore, control experiments with $MoS_2$ flake covered completely by the Au electrodes showed no signature of ion intercalation (Supplementary Fig. 5); this verifies that the Au electrodes can effectively seal the edges of $MoS_2$ and the ions indeed go through the top surface in sealed $MoS_2$. Previous research also showed the metal electrodes could block the entry of $Li^+$ from the edges[14]. Finally, to examine the uniformity of the intercalation from the top surface, we performed a series of Raman measurements on sealed-edge $MoS_2$ flakes (1) on $SiO_2$/Si substrate with the Si Raman peak as reference and (2) on quartz substrate from the backside accessing the bottom surface (Supplementary Fig. 6). We found that the representative Raman peaks of Si substrate were explicitly observed for all the intercalation voltages, which indicates that the excitation laser light has totally penetrated the $MoS_2$ flake and reached Si substrate (Supplementary Fig. 6a). For the intercalated state, the $E_{2g}^1$ and $A_{1g}$ peaks of pristine $MoS_2$ were completely undetectable,

while the intensity of Si peak remained unchanged. Thus, we can confirm that the sealed-edge $MoS_2$ flake was uniformly intercalated through the top surface. We also fabricated the sealed-edge $MoS_2$ device on a transparent quartz substrate and carried out the Raman measurements by illuminating the excitation laser light through the bottom of the substrate directly onto the sealed-edge $MoS_2$ flake. When the sample was intercalated, no intrinsic $E_{2g}^1$ and $A_{1g}$ peaks of pristine $MoS_2$ could be detected, further confirming the uniformity of the intercalation of $MoS_2$ flake from top surface (Supplementary Fig. 6b).

**Selective ion intercalation through the top surface.** To study the ion-selectivity of intercalation through the top surface of the $MoS_2$, we tested intercalation of other alkali ions including $Na^+$ and $K^+$ into $MoS_2$ (Fig. 3). The optical images showed very uniform color change upon $Na^+$ intercalation and de-intercalation for sealed-edge $MoS_2$ (Fig. 3a). In this study, we used $NaCoO_2$ as the counter electrode to provide $Na^+$ source and the OCV of $MoS_2$ vs. $NaCoO_2$ is ~0 V (the potential of $NaCoO_2$ with respect to $Na/Na^+$ is ~3.2 V). When the potential of $MoS_2$ was lowered to ~−2.4 V, the $E_{2g}^1$ and $A_{1g}$ Raman modes shifted and diminished while the new modes ~150 and 200 $cm^{-1}$ emerged (Fig. 3e). Intercalation of $Na^+$ at ~−2.4 V vs. $NaCoO_2$ here (corresponding to ~0.8 V vs. $Na/Na^+$) is believed to fall on the second discharge plateau[28]. When the potential of $MoS_2$ was increased back to ~−1.5 V vs. $NaCoO_2$, the Raman spectrum changed back to that of pristine $MoS_2$. In contrast to the intercalation of $Li^+$, the changes in both color and Raman spectra were also observed to be reversible for the open-edge configuration (Fig. 3c, f). The consistent observation of reversible changes in the color and Raman spectra (analogous to $Li^+$ intercalation (Fig. 2)) demonstrated successful intercalation of the $Na^+$ ion from the top surface into the sealed-edge $MoS_2$. The relatively uniform intercalation of $Na^+$ into open-edge $MoS_2$ was surprising because $Na^+$ (~1.16 Å) is larger than $Li^+$ (~0.9 Å) in size. Based on the comparison of Raman spectra (Fig. 2d and Fig. 3f), this is probably because unlike $Li^+$, the intercalation of $Na^+$ did not induce a large structural deformation. In addition, we note that among all alkali metals, Na has relative weak chemical binding to various substrates including $MoS_2$[35].

On the other hand, no noticeable change was observed in the optical images upon the attempt to intercalate $K^+$ into sealed $MoS_2$ (Fig. 3b). The Raman spectra with different potential of $MoS_2$ vs. K metal from OCV (~3.0 V) to as low as 0 V maintained the same except the very slight peak shift due to the gating effect as in the $Li^+$ case, indicating no $K^+$ intercalation into the sealed $MoS_2$ (Fig. 3g). We suspect that this is due to the large size of $K^+$ (~1.52 Å) compared with $Na^+$ and $Li^+$. In contrast to the sealed-edge configuration, the intercalation of $K^+$ into open-edge $MoS_2$ occurred when we lowered the potential of $MoS_2$ vs. K metal from OCV (~3.0 V) to ~1.1 V (Fig. 3d). This is as expected because the large interlayer spacing (~0.615 nm) of $MoS_2$ can accommodate K$^+$ ions (~1.52 Å) when they intercalate from the edges. Still, for this open-edge $MoS_2$, the color showed a non-uniform change and only partially recovered to its original state, as in the case of $Li^+$ intercalation. In situ Raman spectra showed shift and diminish of the $E_{2g}^1$ and $A_{1g}$ Raman modes and appearance of the new Raman modes ~140 and 190 $cm^{-1}$ after intercalation (Fig. 3h), which was consistent with the observations in the optical images. After de-intercalation, the $E_{2g}^1$ and $A_{1g}$ Raman modes shifted back with a drastically reduced intensity compared

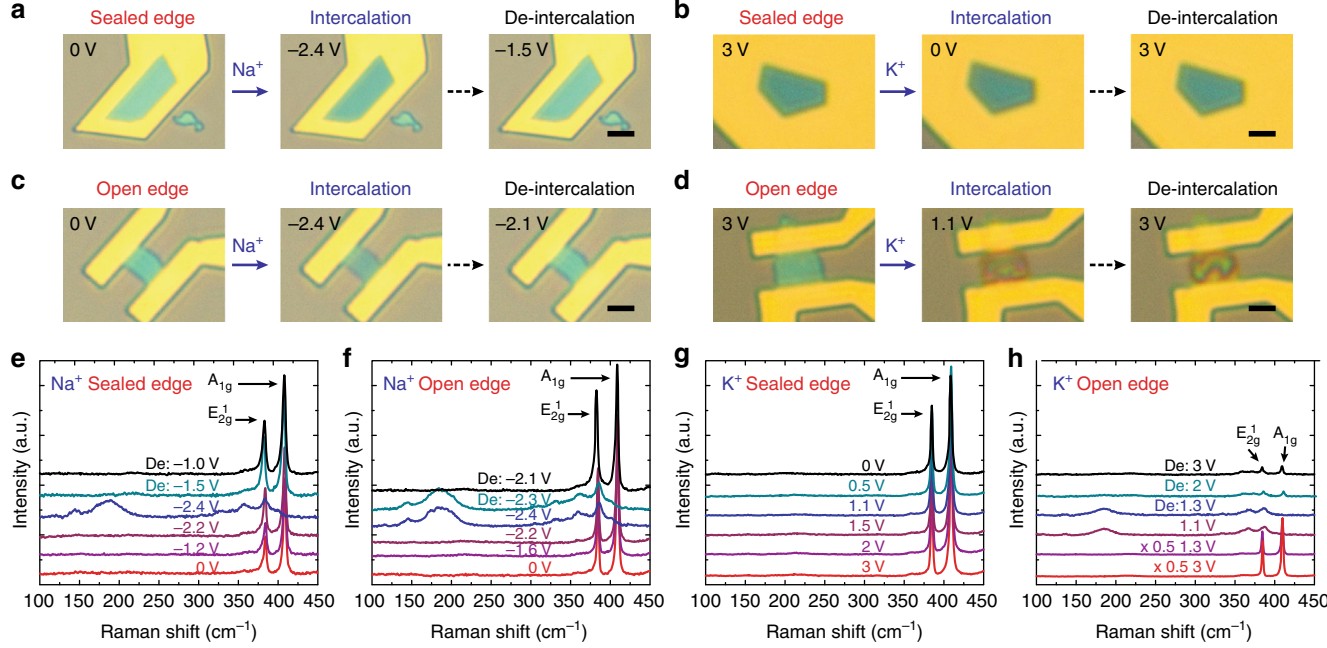

**Fig. 3** Selective intercalation from the top surface of sealed-edge $MoS_2$. **a** In situ optical microscopy images show uniform color change of $MoS_2$ upon $Na^+$ intercalation and de-intercalation for sealed-edge $MoS_2$ through the top surface. **b** The color of $MoS_2$ remains unchanged when lowering the potential from 3 V to 0 V, indicating that $K^+$ cannot intercalate through the top surface of sealed-edge $MoS_2$. **c-d** In situ optical microscopy images show prominent color changes due to $Na^+$ and $K^+$ intercalation through the edges of $MoS_2$. Scale bars in **a-d**, 5 μm. **e-f** In situ Raman spectra demonstrate $Na^+$ intercalates into both sealed-edge $MoS_2$ and open-edge $MoS_2$. **g-h** In situ Raman spectra demonstrate $K^+$ does not intercalate into sealed-edge $MoS_2$ but intercalate into open-edge $MoS_2$. The voltages in **a**, **c**, **e**, and **f** were with respect to $NaCoO_2$. The voltages in **b**, **d**, **g**, **h** were with respect $K/K^+$

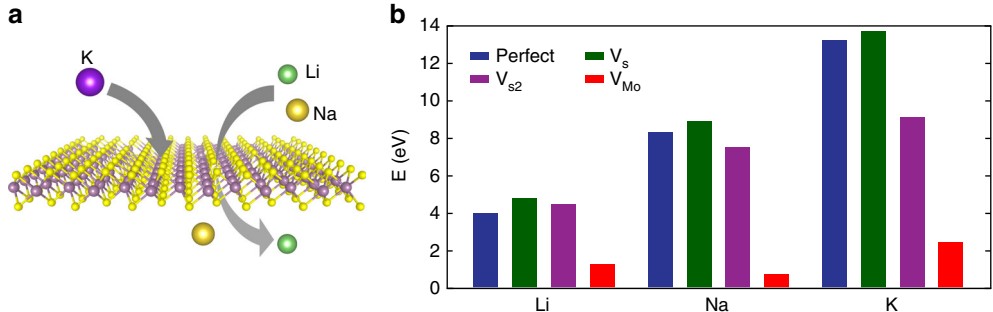

**Fig. 4** DFT calculations for alkali ions penetration through $MoS_2$. **a** Schematic representation of alkali–ion intercalation through a single $MoS_2$ layer. The intercalation pathways allow the penetration of $Li^+$ and $Na^+$ while block $K^+$. **b** Energy barriers for penetration through perfect $MoS_2$ (navy blue) and $MoS_2$ with $V_S$ (dark green), $V_{S2}$, (purple) and $V_{Mo}$ (red) for $Li^+$, $Na^+$, and $K^+$, respectively

with that of the pristine $MoS_2$. In addition, we show that $K^+$ intercalated from the open edges and then extended into sealed areas of the same flake, which provide further evidence that $K^+$ does not intercalate from the top surface of $MoS_2$ (Supplementary Fig. 7).

**Analysis of intercalation pathways through the top surface.** Our results have demonstrated that $Li^+$ and $Na^+$, but not $K^+$, can be successfully intercalated into sealed-edge $MoS_2$ through the top surface. To uncover the underlying mechanism, we propose that $Li^+$ and $Na^+$ intercalate through the natural defects[36,37] into sealed-edge $MoS_2$. We test our hypothesis by comparing the energy barriers for alkali ions to penetrate a monolayer $MoS_2$ with and without defects using density functional theory (DFT) calculations (Fig. 4a). In experiment, an initial applied potential drives alkali ions to accumulate on the top of $MoS_2$ surface, and

these alkali ions maintain their ionic state during the intercalation and de-intercalation. In addition, due to the inversion symmetry of monolayer $MoS_2$, the kinetics dominate over the thermodynamics in the process of intercalation. Therefore, the nudged elastic band method that is widely used to study the kinetic effect in electrochemical reaction[38] is employed here. In previous research, several types of intrinsic point defects have been studied in monolayer $MoS_2$ both theoretically and experimentally, including the single S vacancy ($V_S$), the double S vacancy ($V_{S2}$), single Mo vacancy ($V_{Mo}$), and etc[39]. We first calculated the formation energy for all types of defects and found that $V_S$, $V_{S2}$, and $V_{Mo}$ have relatively low formation energies (Supplementary Fig. 8), which are consistent with previous results[39]. Therefore, we only considered these three types of intrinsic defects and calculated the energy barriers and diffusion pathway for the intercalation of alkaline ions ($Li^+$, $Na^+$, $K^+$) through these intrinsic defects systematically. Intercalation of alkaline ions through the

perfect $MoS_2$ was also considered for comparison. Figure 4b summarizes the energy barriers for $Li^+$, $Na^+$, and $K^+$ to penetrate through perfect $MoS_2$ and $MoS_2$ with $V_S$, $V_{S2}$, and $V_{Mo}$ vacancies, respectively. In the perfect $MoS_2$ monolayer, the energy barriers were 4.03 eV, 8.32 eV, and 13.22 eV for intercalation of $Li^+$, $Na^+$, and $K^+$ through the top surface, respectively. In the presence of $V_S$ and $V_{S2}$ vacancies, these values did not change significantly. In contrast, the energy barriers were significantly reduced when $Li^+$, $Na^+$, and $K^+$ penetrate through the $MoS_2$ monolayer with Mo vacancy, which were 1.30 eV, 0.79 eV, and 2.46 eV, respectively. These energy barriers for $Li^+$ and $Na^+$ to penetrate through $MoS_2$ is comparable with the potential change (difference between OCV and intercalation voltage) in our experiments (1.30 eV vs. ~ 2.2 V and 0.79 eV vs. ~ 2.4 V). In contrast, the energy barrier for $K^+$ to go through $V_{Mo}$ is much larger (2.46 eV) compared with $Li^+$ and $Na^+$, explaining the unsuccessful intercalation of $K^+$ even with ~3.0 V potential change in experiment. Therefore, we believe that the Mo vacancy plays an important role when the alkaline ions intercalate into $MoS_2$ through the top surface (see Supplementary Figs. 9–15 for details).

We checked our $MoS_2$ flakes using high-angle annular dark-field scanning tunneling electron microscopy (HAADF-STEM) and were able to observe strongly reduced brightness at certain Mo sites, which probably indicate the presence of Mo vacancies (Supplementary Fig. 16). Previous scanning tunneling microscopy study has also reported Mo-like vacancies in the $MoS_2$ flakes[36]. Moreover, comprehensive investigations on the defects of monolayer $MoS_2$ through high-resolution STEM have elucidated that the density of $V_{Mo}$ is ~0.004 $nm^{-2}$(ref [40]). In contrast, the dominant sulfur vacancies and disulfur vacancies were reported to have higher densities (density of $V_s$ ~0.12 $nm^{-2}$ and $V_{S2}$ ~0.017 $nm^{-2}$)[40]. Besides the vacancy defects mentioned above, vacancy complex of Mo and three close-by sulfur atoms or disulfur pairs ($V_{MoS3}$ or $V_{MoS6}$) were occasionally observed in $MoS_2$, but the density of $V_{MoS6}$ was much lower than that of $V_{Mo}$, and $V_{MoS3}$ was too low to be counted[40]. We include both the energy barrier for intercalation and the density of the vacancies in our analysis. Considering the energy barriers for the intercalation of $Li^+$, $Na^+$, and $K^+$ through $V_S$ ($V_{S2}$) are, respectively, 4.03 eV (4.51 eV), 8.32 eV (7.53 eV), and 13.22 eV (9.14 eV) and those through $V_{Mo}$ are, respectively, 1.30 eV, 0.79 eV, and 2.46 eV, the relatively lower energy barriers of $V_{Mo}$ are the determinant factor to induce the intercalation through top surface, although the density of $V_{Mo}$ is lower than $V_S$ and $V_{S2}$. In the case of $V_{MoS6}$, however, the extremely low density may dominate and make the intercalation unlikely, although the calculated energy barriers for the intercalation of $Li^+$, $Na^+$, and $K^+$ through $V_{MoS6}$ are only 0.62 eV, 0.65 eV, and 1.24 eV, respectively (Supplementary Fig. 17). This argument is supported by our experimental observation that K ions are always rejected by the surface intercalation pathways. Therefore, we believe the most preferential intercalation pathway through top surface is from the $V_{Mo}$. Furthermore, we would like to emphaize that the real situation in the experiments is more complicated. First, the $V_S$, $V_{S2}$, and $V_{Mo}$ may co-exsit and the S vacancies can lead to formation of Mo vacancies; second, the vacancies can evolve during intercalation due to the insertion of the ions. Nevertheless, the DFT calculations confirm that it is feasible to intercalate through the top surface into the few-layer $MoS_2$ and the intercalation has selectivity.

**Reversible control of optical and electrical properties**. Both the in situ optical microscopy and Raman spectroscopy results show the high reversibility and stability of the intercalation from the top surface of the few-layer $MoS_2$. Finally, we demonstrate the reversible control of both optical and electrical properties of few-layer $MoS_2$ as a first step toward applications in optoelectronics and nanoelectronics devices (Fig. 5). Since $Li^+$ intercalation of 2D $MoS_2$ has been well-studied[13,14], we focused on $Na^+$ intercalation here. We first measured the reflectance spectra of the $MoS_2$ flakes upon $Na^+$ intercalation, which can also quantify the subtle color change in the optical images (Fig. 5a). For the pristine flake, two reflectance dips were observed at ~1.86 eV and 2.02 eV (red line at bottom, Fig. 5b), as a result of enhanced absorption in the $MoS_2$ by the A and B excitons. These well-characterized excitons correspond to the prominent transitions between the maxima of split valence bands and the minimum of the conduction band, located at the K point of the Brillouin zone[41,42]. As we gradually lowered the potential of $MoS_2$ vs. $NaCoO_2$ to −2.6 V, excitonic transition B exhibited a minor blue-shift and a clear damping in intensity, due to the decrease in exciton binding energy resulted from the doped-free electrons[43,44]. The change in excitonic transition A also showed indication of damping upon ion intercalation, while this was more difficult to identify because it overlapped with a broad background Fabry–Pérot resonance in the $MoS_2$–$SiO_2$–Si layer stack (Supplementary Fig. 18). We performed analytical transfer-matrix calculations of the experimental geometry to resolve the Fabry–Pérot resonance besides both excitonic transitions (Supplementary Fig. 18). When we increased the potential of $MoS_2$, the spectral shifts and intensity change are both fully reversed. To the best of our knowledge, this is the first report that optical properties of $MoS_2$ were electrically manipulated via $Na^+$ intercalation, which shows reversible and reproducible tuning over multiple cycles (Fig. 5c). Identical observations were found in the case of $Li^+$ intercalation (Supplementary Fig. 19).

Next, we performed in situ electric transport measurements to show the highly tunable electrical properties via ion intercalation. We note that it is challenging to find an insulating material with high malleability, electrochemical stability, and high affinity to seal $MoS_2$ edges for the electrical measurements at this point (Supplementary Fig. 20). Since $Na^+$ intercalation through top surface and edges were equally reversible (Fig. 3), we employed open-edge configuration to simplify the device geometry. For the two-contact device shown in Fig. 3c, the drain-to-source current increased dramatically when the potential of $MoS_2$ was continuously swept from 0 V to −2.6 V with respective to $NaCoO_2$ counter electrode (Fig. 5d), because of the intercalation of $Na^+$ ions into $MoS_2$. When the potential was returned to around −2.0 V, the current dropped rapidly due to the extraction of $Na^+$ ions from $MoS_2$. Besides the current hysteresis between −1.6 V to −2.6 V arising from the over-potential effect, the current curves nearly overlapped with each other in the non-intercalated state (from 0 V to −1.6 V), which implied the restored semiconducting phase of de-intercalated $MoS_2$. Moreover, the current consistency of three cycles demonstrated the stability of electrical properties after $Na^+$ ion intercalation. To eliminate the effect of contact resistance and study the intrinsic transport properties, we fabricated another device with standard Hall-bar geometry and changed the applied potentials only at 300 K, higher than the ineffective temperature of electrolyte (Supplementary Fig. 21). The four-probe resistivity (Fig. 5e) reduced exponentially till nearly saturated due to the surface charging from electric-double-layer effect when the potential of $MoS_2$ was continuously swept from −0.8 V to −2.3 V. As the sample was cooled down, all the resistivity decreased and displayed metallic behavior down to 2 K. When the $Na^+$ ions were intercalated at the potential of −2.6 V, the resistivity decreased by an order of magnitude compared with that prior to the intercalation at −2.3 V. From the Hall effect measurements (Supplementary Fig. 22), we found that the electron density at

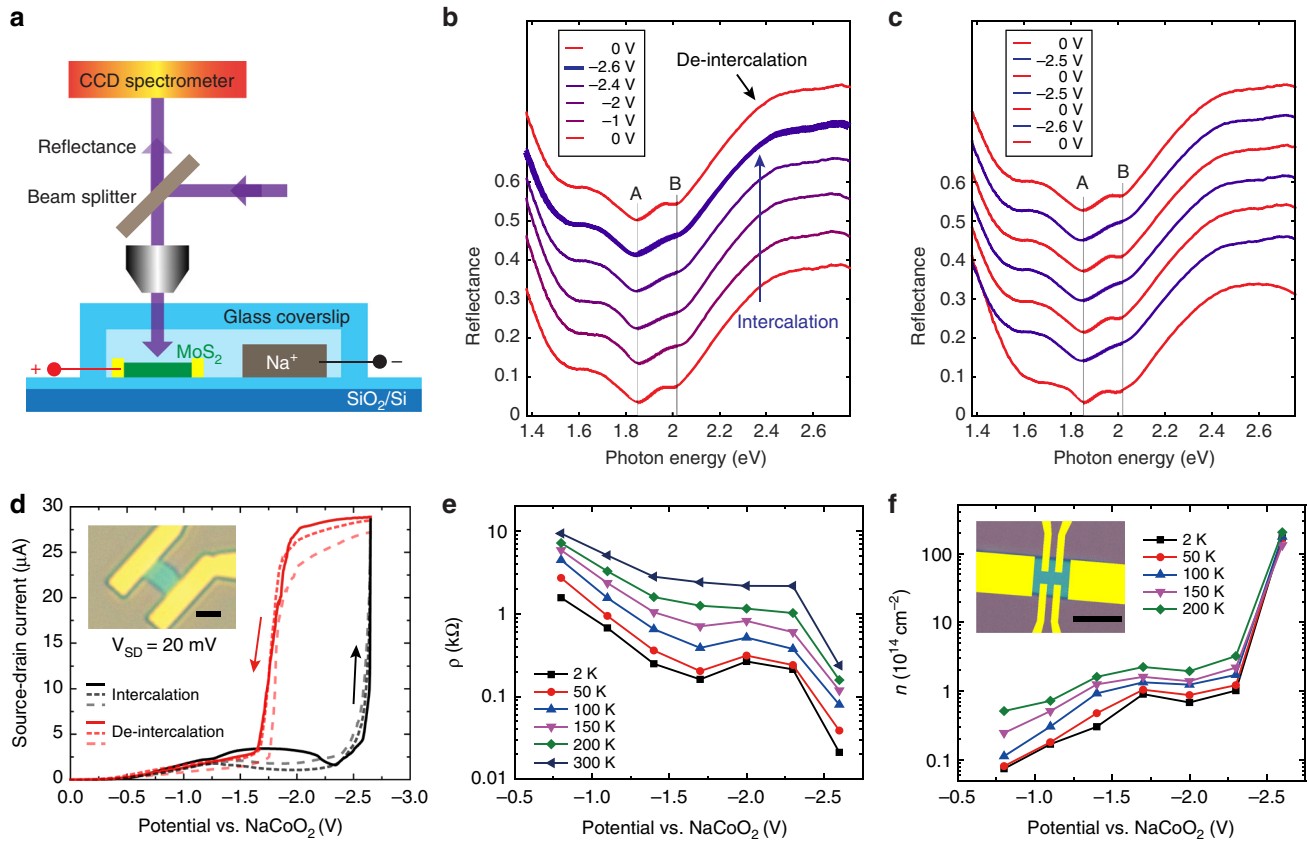

**Fig. 5** Reversible control of optical and electrical properties of MoS$_2$ via Na$^+$ intercalation. **a** Schematic of the optical measurement. **b** Gradual reflectance change of MoS$_2$ on SiO$_2$/Si via Na$^+$ intercalation (spectra offset for visibility). From bottom red line to middle blue line: Na$^+$ intercalation; top red line: Na$^+$ de-intercalation. **c** Three cycles of reflectance change of MoS$_2$ on SiO$_2$/Si via Na$^+$ intercalation. **d** The potential dependence of source-drain current within three intercalation/de-intercalation cycles. The bias between source and drain was V$_{SD}$ = 20 mV at room temperature. The potential of MoS$_2$ was swept from 0 V to −2.6 V and back to 0 V (vs. NaCoO$_2$ counter electrode) for three cycles at a constant rate of 2 mV s$^{-1}$. The black and red arrows indicate the sweeping directions. The inset shows the device. Scale bar, 5 μm. **e** The resistivity of MoS$_2$ flake (5.5 nm) with standard Hall-bar geometry. The potential of MoS$_2$ was applied at 300 K. The device was then continuously cooled down to 2 K with fixed voltages. **f** The corresponding two-dimensional electron densities at different potentials and temperatures. The inset shows the device used in (**e**) and (**f**). Scale bar, 5 μm

−2.3 V can reach 1×10$^{14}$ cm$^{−2}$ at 2 K (Fig. 5f), consistent with that in ionic-liquid gated MoS$_2$[9,45]. After Na$^+$ intercalation, the electron density was increased up to 1.7×10$^{16}$ cm$^{−2}$, which was two orders higher than those reported using dielectric gating or liquid gating[8,9]. In 3D, it corresponds to 3×10$^{22}$ cm$^{−3}$ and Na$_{1.6}$MoS$_2$ in molecular formula by assuming one Na atom contributed one electron to the rigid MoS$_2$ host. To the best of our knowledge, this is the highest charge-carrier density ever achieved in doped MoS$_2$. Here, we use Na$^+$ intercalation in open-edge MoS$_2$ to show highly tunable and reversible electrical properties through ion intercalation, and the results can be extrapolated to Li$^+$ or Na$^+$ intercalation in sealed-edge configurations which should show even more stable performances. The reversible electrochemical control of the optical and electric performances of few layer MoS$_2$ opens a new route to design highly tunable and stable 2D material-based optoelectronic and nanoelectronic devices.

## Discussion

In summary, we demonstrate that Li$^+$ and Na$^+$ ions can intercalate into few-layer MoS$_2$ through the top surface with strongly improved control, reversibility, and stability compared with intercalation through edges. We note intercalation through top surface also applies to other similar 2D layered material systems such as MoSe$_2$ (Supplementary Fig. 23). This finding is significant

because electrochemical control is a powerful approach to manipulate the properties of low-dimensional materials; stable intercalation and reversible cycling are essential for accurate in situ interrogation of the physical and chemical changes during intercalation. In future, sealing the edges of 2D materials with dielectrics will allow the design of complex and tunable nanoelectronic devices with high performance and high stability. In addition, voltage-controlled selective intercalation through the top surface of the 2D materials holds great potential in developing novel ionic sieving devices that have distinct open/close (on/off) controllability via a gate voltage applied on the 2D material, besides the size and charge selectivity from common nanofiltration and desalination membranes.

## Methods

**Device fabrication.** Thin MoS$_2$ flakes (<10 nm) were exfoliated using the Scotch tape method onto 300-nm-thick SiO$_2$/Si substrate, and electrodes (Ti/Au, 3/50 nm) were designed and patterned on flakes by electron-beam lithography and deposited by e-beam evaporation. A marker array was used for precise alignment of the electrodes to the selected flakes. The MoS$_2$ sample was then transferred to an Ar-filled glovebox for cell assembly. In the case of Li$^+$ and K$^+$ intercalation, Li/K metal was cold pressed onto Cu foil as the counter electrode; in the case of Na$^+$ intercalation, NaCoO$_2$ was deposited onto an Al foil as the counter electrode. These electrodes were then sealed between a cover glass and the SiO$_2$/Si substrate with evaporated Ti/Au electrode, using hot melt sealing film (Meltonix 1170–60, Solaronix), leaving two little openings for liquid electrolyte filling. There is a ~50 μm gap between the glass and the SiO$_2$/Si substrate, which is then occupied by the corresponding electrolytes. The electrolytes were 1 M LiPF$_6$, NaPF$_6$, KPF$_6$ in EC/

DEC for Li+, Na+, and K+ intercalation, respectively. After filling the electrolyte by capillary effect, the two openings were sealed by using epoxy.

**Electrochemical intercalation**. Electrochemical intercalation was performed with a Keithley 2400 sourcemeter. Constant voltage charge and discharge were used to intercalate and de-intercalate MoS2 flakes for in situ optical and Raman measurements.

**Transport measurements**. The room temperature current between drain and source was recorded by using the Keithley 2400 sourcemeter, while another Keithley 2400 sourcemeter was used to apply charge and discharge voltage between the source and counter electrode. The low temperature transport measurements were carried out in Quantum Design PPMS-7 instrument, Janis 9 T magnet He-cryostats (base temperature 2 K), using low-frequency (5–20 Hz) AC technique by digital lock-in amplifiers (Stanford Research Systems SR830) with current-driven configuration. The charge-carrier densities were derived from Hall effect measurements.

**Raman spectroscopy**. The MoS2 flakes were characterized using HORIBA Scientific LabRAM HR Evoluation spectrometer, with 532 nm excitation and 1800 l/mm grating. The background signals from electrolyte, cover glass, and substrates were subtracted.

**Optical measurement and simulations**. Optical reflection spectra were measured using a Nikon C1 confocal microscope. Unpolarized broadband excitation from a halogen lamp and a 50 × long working distance objective (NA = 0.55) were used to illuminate the sample through the cover slip. A 30-μm pinhole was used to spatially select the reflection off the flakes, and a Princeton Instruments SpectraPro 2300i (150 l/mm, blazed at 500 nm) and PIXIS CCD camera were used to measure the spectrum. A protected silver mirror (Thorlabs) was used as a reference to correct for the system response.

**DFT calculations**. The spin-polarized density functional theory (DFT) calculations were implemented in the Vienna Ab-initio Simulation Package (VASP)[46,47]. The projected augmented wave[48,49] and Perdew–Burke–Ernzerhof (PBE) functional[50,51] were used to describe the electron–ion interaction and exchange-correlation energy, respectively. The cutoff energy was set to 500 eV. Because of the layered structures of MoS2, the empirical correction method proposed by Grimme (DFT-D3) was used to simulate the van der Waals interaction[52,53]. In order to understand the intercalation process, a supercell consisting of 5×5 repeating unit cells of MoS2 monolayer was used, and a vacuum layer was larger than 15 Å to eliminate the spurious interaction between adjacent MoS2 layers. 3 × 3 × 1 Γ-centered k-points was used to sample the Brillouin zones of supercell. For each model, the structure was allowed to fully relax until the energy converged to 10$^{-5}$eV and the residual force converged to 0.01 eV/Å per atom. The climbing image nudged elastic band (CI-NEB) method[54] was applied to search for the transition state and determine the migration barrier of alkaline ions through the MoS2 monolayer.

## Data availability
The data that support the findings of this study are available from the corresponding author upon reasonable request.

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

## Acknowledgements

This work was supported by the Department of Energy, Office of Basic Energy Sciences, Division of Materials Sciences and Engineering (contract no. DE-AC02–76SF00515). This work was also supported by the National Key Research and Development Program of China (Grant Nos. 2017YFB0701600 and 2014CB932400), the National Nature Science Foundation of China (Grant No. 11874036) and Shenzhen Projects for Basic Research (Grant No. JCYJ20170412171430026). Tianjin Supercomputing Center is acknowledged for allowing the use of computational resources including TIANHE-1. J.G. and M.L.B. acknowledge support from the Department of Energy Grant DE-FG07-ER46426. Part of this work was performed at the Stanford Nano Shared Facilities (SNSF), supported by the National Science Foundation under award ECCS-1542152. J.Z. and A. Y. acknowledge helpful discussions with Dr. Xiao-Xiao Zhang.

## Author contributions

J.Z., A.Y., and Y.C. designed research. J.Z., A.Y. fabricated devices and performed optical microscopy and Raman spectroscopy measurements. X.W., P.T., J.L., and S.C.Z. performed DFT calculations and theoretical analysis. J.G., A.Y., Q.L., and M.L.B performed optical measurement and simulations. J.Z. and S.L. performed transport measurements. B.L., F.S., J.W., Y.S., Z.L., G.Z., and C.-L.W. contributed to the sample fabrication and processing. X.Z. performed STEM measurement. A.Y., J.Z., P.T., J.G., and Y.C. analyzed the data and wrote the paper. All authors participated in discussions.

## Additional information

**Competing interests:** The authors declare no competing interests.

