## [Peer Review File · Nature Communications]

Parts of this Peer Review File have been redacted due to copyright infringement.

Reviewers' comments:

Reviewer #1 (Remarks to the Author):

It is an interesting idea, from usually creative yet thorough research group of Yi Cui. But the thoroughness seems to be bit compromised, probably zealous first author! Here are some comments/queries.

1) In graphene, Li ion transport through the top surface is attributed to defects in the structure. This is discussed here in MoS₂ via DFT calculations. One would think there would be some analysis of the defect density in the MoS₂ of the various defects discussed, which is sorely missing. There is no major postmortem analysis to fully justify the assertions.

2) Just a few cycles of an intercalation-based reaction doesn't really show 'robustness' of the material, especially since there is noticeable deterioration in the Raman signal. Many other materials that have lithium intercalation are able to be cycled for much longer and number. So its unclear how valid some of the stability claims are.

3) The authors indicate that the mechanical stabilization of the flake contributed to reversibility, in addition to the 'uniform distribution' of diffusion pathways of the surface (pg 7), so it would be curious if authors are able to determine which is more important or has the larger effect as one is an extrinsic effect and the other is an intrinsic property of the material.

4) do the authors know what the morphology/interface of the MoS₂/Au sealed edge look like after Au deposition (via AFM, etc?).

5) is the same degree of Li intercalation achieved in both the open and sealed edge configuration?

Finally, authors attribute better cycling stability to the surface and geometry of the electrode, which makes it difficult to accept that the top surface is better for intercalation than the edges, if indeed other factors come into play.

Overall, it is an interesting idea and a bold assertion. However, the experimental data just does not justify the claims. It may indeed be valid and correct but as presented there is just not "there there".. Major characterization efforts are necessary and lot more cycling and comparisons would be warranted.

Reviewer #2 (Remarks to the Author):

The authors successfully demonstrated reversible ion intercalation through top surface of few-layer MoS₂ with all edges sealed and confirmed that the top-surface intercalation is more reversible and stable compared to edge intercalation. The authors further investigated the optical and transport properties of the intercalated MoS₂. These results are helpful for developing optoelectronic and nanoelectronics devices and deserve publication. However, the following issues have to be resolved before acceptance (Minor Revision).

1. The uniformity of top-surface intercalated sample in thickness direction. The top-surface intercalation requires more energy, could it be able to uniformly intercalate the sample? How long it may take to fully intercalate the sample? Such information is important, the author should confirm the uniformity of the sample. For example, please provide the surface sensitive optical properties of the bottom surface (making device on a cover glass).

2. Because the top-surface intercalation is based on the existence of natural defects in exfoliated MoS₂ flake, is it applicable to other 2D materials?

3. The ion intercalation rate (charging rate) also plays an important role in reversibility and

uniformity of the device, please discuss for both top-surface and edge intercalation.

4. Because this work focuses on top-surface intercalation, to highlight its advantage, please provide the transport properties of the top-surface intercalated devices, instead of the edge-intercalated devices. Comparing their difference is also helpful. This is helpful but not required.

5. Please provide the Hall data (Hall voltage vs Magnetic field at respective potentials and temperatures) in the supplementary information. The potential was applied at 200K, why the carrier density changes as a function of temperature at fixed potential (the electrolyte was frozen at low temperature)?

Reviewer #3 (Remarks to the Author):

The authors report on electrochemical ion intercalation as a potential materials synthesis route for tunable optoelectronic, energy and separation properties of 2D materials. They demonstrate this with MoS₂ as an example case where they show reversible intercalation with sealed edges and argue based on a suite of experimental and DFT calculations that smaller cations can "intercalate" while larger cations cannot through the top surface. The manuscript is reasonably well-presented and the findings are certainly intriguing. However, there are serious issues that need to be addressed first.

1) The use of "plain vanilla" nudged elastic band type calculations to discuss barriers in the context of an electrochemical intercalation reaction is deeply flawed. This is because the work function of the system needs to be held constant and electrochemical reaction barriers simply cannot be estimated in the way done in Figs S6-S10 as the trajectory taken by NEB does not guarantee constant electrochemical potential. Please see for e.g., *Chemical Physics Letters*, 466, 68-71, *J. Phys. Chem. C*, 112 (2008), pp. 8747-8750, *Phys. Rev. B*, 73 (2006), p. 115407, etc. While they may have some data value, they provide no direct comparison to the experimental measurements carried out in this work.

2) The authors need to microscopy write down how the process is occurring: where is the electron transfer occurring?

3) How about the thermodynamics of electrochemical ion insertion for these different processes? How does that compare to the experimental numbers? Is it a thermodynamic vs kinetic effect? The authors should estimate this through the standard computational lithium and sodium electrode scheme for concerted ion-coupled electron transfer processes.

4) "These diffusion energy barriers through VMo are on the same order of magnitude as the applied energy in the experiments (difference between OCV and applied voltage)." I am not sure I am convinced of this statement at all, in light of comments above. Also, authors should clarify make the distinction between thermodynamics and kinetics in this case.

5) How many cycles was it possible to reversibly demonstrate the insertion/de-insertion?

REVIEWER #1

Comment: It is an interesting idea, from usually creative yet thorough research group of Yi Cui. But the thoroughness seems to be bit compromised, probably zealous first author! Here are some comments/queries.

Our Response:

We thank Reviewer 1 for the highly positive evaluation of our work and the creativity of our group. The questions and suggestions raised by Reviewer 1 are extremely important and helpful for improving the quality of our work. As will be shown below, we have carried out a series of new measurements following Reviewer 1's comments. The main conclusion of our manuscript is further strengthened, and we believe the quality of the paper is significantly improved.

Comment: In graphene, Li ion transport through the top surface is attributed to defects in the structure. This is discussed here in MoS₂ via DFT calculations. One would think there would be some analysis of the defect density in the MoS₂ of the various defects discussed, which is sorely missing. There is no major postmortem analysis to fully justify the assertions.

Our Response:

We thank Reviewer 1 for bringing up these questions, which we should have clarified more thoroughly in our original manuscript.

(1) About the defect analysis

In the mechanically exfoliated MoS₂, there are various types of defects, including sulfur vacancy (V_S), disulfur vacancy (V_{S_2}), molybdenum vacancy (V_{Mo}), vacancy complex of Mo and three close-by sulfur (V_{MoS_3}), vacancy complex of Mo and three close-by disulfur pairs (V_{MoS_6}), and antisite defects where a S atom or S₂ column substituting a Mo atom (S_{Mo} or S_{2Mo}) and a Mo atom substituting a S atom or S₂ column (Mo_S or Mo_{S_2}). It has been widely accepted that the dominant defects are S vacancies (Nat Comm 6: 6293 (2015), now ref 40, and ref 36, 37, 39), and the minor vacancies are V_{S_2} , V_{Mo} , V_{MoS_3} , V_{MoS_6} . Since the antisite defects are unlikely to enable the ion intercalation, we would not discuss these defects in the following content.

To clearly identify the defects, we performed high-angle annular dark-field scanning transmission electron microscopy (HAADF-STEM) imaging to reveal Mo vacancies on our exfoliated MoS₂ flakes (Figure R1). The red circles indicate the defects with much reduced brightness at Mo sites of monolayer MoS₂ (Figure R1a) and few-layer MoS₂ (Figure R1b), which probably arise from the V_{Mo} . Due to low resolution of our STEM, we cannot explicitly rule out the possibility of S_{Mo} contribution to the reduced intensity at Mo sites. However, previous scanning tunneling microscopy (STM) study (ACS Appl. Mater. Interfaces 7, 11921–11929 (2015), ref 36) has reported the Mo-like vacancies in the MoS₂ flakes. Moreover, comprehensive investigations on the defects of monolayer MoS₂ through high-resolution STEM (Nat. Comm., 6, 6293 (2015), ref 40) have elucidated that the density of V_{Mo} is around 0.004 nm⁻² (as shown in Figure R2-a). And the dominant sulfur vacancies and disulfur vacancies were reported to have higher densities (density of $V_S = 0.12$ nm⁻² and $V_{S_2} = 0.017$ nm⁻²) in the same report (Figure R2-a). In addition, the density of complex vacancy of V_{MoS_6} is much lower than that of V_{Mo} , and V_{MoS_3} is too low to be counted (as shown in Figure R2-a).

Considering the energy barriers for the intercalation of Li^+ , Na^+ and K^+ through V_s are respectively 4.03 eV, 8.32 eV and 13.22 eV and that through V_{Mo} are respectively 1.30 eV, 0.79 eV and 2.46 eV, the relatively lower energy barriers of V_{Mo} are the determinant factor to induce the intercalation through top surface when reducing the applied potentials, although the density of V_{Mo} is about 30 times lower than V_s . Therefore, the most preferential intercalation pathway through top surface is from the V_{Mo} , especially for Na^+ ions. However, the energy barrier for K^+ ion through the V_{Mo} is much higher than the other two ions, indicating the lower probability of K^+ intercalation through top surface.

For completeness, we have also calculated the energy barrier for the intercalation of Li^+ , Na^+ and K^+ through V_{MoS_6} , which are 0.62 eV, 0.65 eV and 1.24 eV, respectively (Figure R2-c). Although the energy barrier for the intercalation of K^+ is twice as large as that of Li^+ and Na^+ , these energy barriers are much smaller than the potential change (difference between OCV and intercalation voltage) in experiment (~ 2.2 - 2.4 V), indicating that they would be easily overcome. However, (1) the much higher formation energy of V_{MoS_6} than V_{Mo} (See Figure 2c: Formation energy for different defects from Nano Letters 13, 2615–2622 (2013)), leads to much lower densities of V_{MoS_6} than V_{Mo} , which is verified by the STEM (Figure R2-a, b); (2) experimentally, we observed that K ions were rejected by the surface intercalation pathways. Therefore, we believe the extremely low density of V_{MoS_6} makes it difficult to observe any intercalated signal and the V_{Mo} paths are the most reasonable explanation in our experiment.

Figure R1 HAADF-STEM images of exfoliated MoS_2 , the red dashed circles indicate Mo vacancies. (*This figure is now included as Figure S16*)

Figure R2 Histograms of various point defects in PVD, CVD and mechanically exfoliated (ME) monolayer MoS₂, adapted from Nat Comm 6: 6293 (2015), ref 39 (*This figure is for reviewers only*)

[redacted]

Figure R2-c The migration of Li^+ , Na^+ and K^+ through V_{MoS_6} (This figure is now included as Figure S17)

(2) About the postmortem analysis

The best way to perform postmortem analysis is through TEM characterization of intercalated MoS_2 . However, this approach would involve mechanically breaking the cell, rinsing with non-polar solvents, and PMMA-assisted transferring to TEM grid. These steps will expose MoS_2 to air, solvents, KOH etchant, etc., which will complicate the analysis or even damage the MoS_2 flakes. We have not successfully performed such experiments on electrochemically intercalated 2D materials yet. Instead, by visualizing the defects before intercalation from STEM as well as extracting useful (and more expert) defect analyses in past reports, we provide a detailed discussion on the defects (added to main text). This discussion, together with the experimental demonstration of clear changes in *in-situ* Raman spectra, optical microscopy images, optical and electrical properties, and our comprehensive DFT calculations, show that intercalation from the top surface is likely induced by the existence of V_{Mo} .

Change to Manuscript: We have discussed the defect densities in the main text as follows:

Page 12:

“We checked our MoS_2 flakes using high-angle annular dark-field scanning tunneling electron microscopy (HAADF-STEM) and were able to observe strongly reduced brightness at certain Mo sites, which probably indicate the presence of Mo vacancies (**Fig. S16**). Previous scanning tunneling microscopy study has also reported Mo-like vacancies in the MoS_2 flakes.³⁶ Moreover, comprehensive investigations on the defects of monolayer MoS_2 through high-resolution STEM have elucidated that the density of V_{Mo} is around 0.004 nm^{-2} .⁴⁰ In contrast, the dominant sulfur vacancies and disulfur vacancies are reported to have higher densities (density of $V_{\text{s}} \sim 0.12 \text{ nm}^{-2}$ and $V_{\text{S}_2} \sim 0.017 \text{ nm}^{-2}$).⁴⁰ Besides the vacancy defects mentioned above, vacancy complex of Mo and three close-by sulfur atoms or disulfur pairs (V_{MoS_3} or V_{MoS_6}) are occasionally observed in MoS_2 , but the density of V_{MoS_6} is much lower than that of V_{Mo} , and V_{MoS_3} was too low to be counted.⁴⁰ We include both the energy barrier for intercalation and the density of the vacancies in our analysis. Considering the energy barriers for the intercalation of Li^+ , Na^+ and K^+ through V_{s} (V_{S_2}) are respectively 4.03 eV (4.51 eV), 8.32 eV (7.53 eV) and 13.22 eV (9.14 eV) and those through V_{Mo} are respectively 1.30 eV, 0.79 eV and 2.46 eV, the relatively lower energy barriers of V_{Mo} are the determinant factor to induce the intercalation through top surface, although the density of V_{Mo} is lower than V_{s} and V_{S_2} . In the case of V_{MoS_6} , however, the extremely low density may dominate and make the intercalation unlikely, although the calculated energy barriers for the intercalation of Li^+ , Na^+ and K^+ through V_{MoS_6} are only 0.62 eV, 0.65 eV and 1.24 eV, respectively (**Fig. S17**). This argument was supported by our experimental observation that K ions are always rejected by the surface

intercalation pathways. Therefore, we believe the most preferential intercalation pathway through top surface is from the V_{Mo} ."

Comment: Just a few cycles of an intercalation-based reaction doesn't really show 'robustness' of the material, especially since there is noticeable deterioration in the Raman signal. Many other materials that have lithium intercalation are able to be cycled for much longer and number. So it's unclear how valid some of the stability claims are.

Our Response:

We thank Reviewer 1 for pointing out this issue.

- (1) Indeed, as the cathode materials in Lithium-ion batteries, MoS_2 or graphite can be cycled many times with excellent cyclability and stability in a good number of publications (for example, in Wan, J, et al, *Advanced Energy Materials* 2015, 5, 1401742). But these cathodes are made of composite slurry coated on metal foils.
- (2) For nano-scale flakes, however, the previous studies on open-edge MoS_2 flakes (Xiong, F, et al. *Nano Letters* 2015, 6777-6784) and on graphite flakes (Bao, W., et al. *Nat. Commun.* 2014, 5, 4224) show much worse cyclability and stability (~ 3 cycles). This is because the nanoflakes is directly exposed to the electrolyte and connected to the Au electrode.
- (3) The most unique and significant advantage of nano-scale flakes is to *in situ* interrogate the physical and chemical properties during the intercalation process. To the best of our knowledge, our sealed MoS_2 flakes are much more stable than the previous studies (Xiong, F, et al. *Nano Letters* 2015, 6777-6784 and Bao, W., et al. *Nat. Commun.* 2014, 5, 4224).

To further demonstrate the stability of sealed MoS_2 , we carried out the Raman measurements up to 20 cycles. As shown in the following figure, both the E_{2g}^1 and A_{1g} peaks can be clearly identified for all the 20 cycles, indicating the crystalline stability of MoS_2 after Li intercalation and deintercalation procedures. Moreover, the 1T phase peaks from 154 to 207 cm^{-1} always show up for the intercalated state. Again, Our Raman results further confirm the stability of sealed MoS_2 flakes.

Figure R3 Raman spectra of Li⁺ intercalation into sealed MoS₂ flakes up to 20 intercalation and de-intercalation cycles. The blue (red) curves represent the Li-deintercalated (intercalated) states. (*This figure is now included as Figure S4*)

Change to Manuscript: We added discussion about the stability up to 20 cycles in the main text as follows:

Page 8:

“In addition, we found both the E_{2g}^1 and A_{1g} Raman modes can be clearly identified for up to the 20 cycles of Li intercalation and de-intercalation processes (**Fig. S4**), indicating the crystalline stability of sealed-edge MoS₂.”

Comment: The authors indicate that the mechanical stabilization of the flake contributed to reversibility, in addition to the ‘uniform distribution’ of diffusion pathways of the surface (pg 7), so it would be curious if authors are able to determine which is more important or has the larger effect as one is an extrinsic effect and the other is an intrinsic property of the material.

Our Response:

We thank Reviewer 1 for raising this question. The surrounding Au electrode indeed provides enough mechanical stabilization of MoS₂ flakes, which prevents structural deformation and guarantee the reversibility. In addition, the “uniformly distributed” diffusion pathways on the top surface are mechanically inextensible and block large organic molecules, leading to more homogeneous intercalation compared to that through the edges where all ions flood into the opening van der Waals gaps. However, these two mechanisms always work together, and we cannot separate them experimentally to determine which one is more important. On one hand, without the surrounding Au electrodes, all the ions would be intercalated through the edges rather than the top diffusion pathways, because the former one has relatively lower intercalation barrier. On the other hand, if the top surface

is a perfect layer without any vacancy defects, no ions can be intercalated, thus surrounding Au electrode cannot come into play to manifest its mechanical stabilization effect. Still, we would believe that the mechanical stabilization from surrounding Au electrode is more important to contribute to the reversibility, in the sense that the diffusion pathways on the top surface are always present in the MoS₂ flakes and can only be triggered when all the edges are sealed. However, we cannot neglect the important contribution from uniformly distributed and mechanically inextensible diffusion pathways on the top surface.

Change to Manuscript: We add more discussions in the main text to make this part clearer to readers.

Page 6-7:

“Although these factors cannot be separated, we believe the mechanical stabilization from surrounding Au electrode contributes more to the reversibility. Because the diffusion pathways on the top surface are naturally present in the MoS₂ flakes, and these can only be triggered when all the edges are sealed.”

Comment: Do the authors know what the morphology/interface of the MoS₂/Au sealed edge look like after Au deposition (via AFM, etc?).

Our Response:

We thank the reviewer for suggesting this experiment. As suggested, we performed the AFM measurements before and after the Au deposition (50 nm) to inspect the morphology/interface of MoS₂/Au sealed edge devices. As shown in the figure below, the Au deposition is mild and uniform, and does not damage the covered MoS₂ surface, resulting in a smooth interface in between. We can clearly identify the step edge of the Au surface which is due to the existence of the flake; therefore, the flake was successfully sealed by the Au deposition.

Figure R4 The optical and atomic force microscope images of MoS₂ flake (a) before and (b) after Ti/Au electrode deposition. (This figure is now included as Figure S1)

Change to Manuscript: We added discussions to describe the morphology and interface of MoS₂ and Au electrode.

Page 5:

“We also performed AFM measurements before and after the Au depositions to inspect the MoS₂/Au morphology and interface. The clear steps of the Au metal on top of MoS₂ edges indicated that the Au deposition was mild and uniform and successfully sealed the MoS₂ flake without damaging the covered MoS₂ surface (Fig. S1).”

Comment: Is the same degree of Li intercalation achieved in both the open and sealed edge configuration?

Our Response:

The best way to probe the degree of Li intercalation is to use the inductively coupled plasma mass spectrometry (ICP-MS), coulometric method, or Hall effect measurements. However, the mass of a single MoS₂ flake is too low to perform ICP-MS measurements. Also, the side electrochemical reaction between Au electrode and the electrolyte restricts the application of coulometric method, because only a tiny amount of change (1.52×10^{-9} C) is already enough to charge a MoS₂ flake ($10 \mu\text{m} \times 10 \mu\text{m} \times 5 \text{nm}$) to Li₁MoS₂ state. Moreover, for the sealed-edge MoS₂, the surrounding Au contact hinders the transverse Hall voltage measurements. Therefore, we can only use the *in-situ* Raman spectra to estimate whether the degree of Li intercalation is on the same level by comparing the peak positions.

As shown in the following figure (Fig. R5), the Raman peaks around 150 and 200 cm⁻¹ associated with 1T phase were towards to lower wavenumbers for the open-edge configuration, implying that the degree of Li intercalation for the open-edge configuration is slightly higher than that of the sealed-edge configuration. This is reasonable considering that all ions can flood into the open-edge MoS₂ with much lower energy barriers. However, from experiments, we found that it is hard to control the degree of Li intercalation in the open-edge configuration (even if we do not lower the potential too much, see our Figure S2 at 1.0 V) and this is also the reason for its irreversibility. In our experiment, our rough criteria for similar degree of Li intercalation is that both configurations arrive at “intercalated state”, *i.e.*, the appearance of the new Raman peaks around 150 and 200 cm⁻¹.

Figure R5 Raman spectra of MoS₂ in intercalated state (0.8 V) for open and sealed edge configurations. (This figure is for reviewers only)

Change to Manuscript: We briefly commented on the degree of Li intercalation in the main text.

“We note that by carefully comparing the peak positions of the Raman modes of 1T phase, we found that the degree of Li intercalation in open-edge MoS₂ is slightly higher than that of the sealed-edge MoS₂. This is due to the lower energy barrier for intercalation in open-edge MoS₂, which makes it hard to control the intercalation process and partially accounts for the irreversibility.”

Comment: Finally, authors attribute better cycling stability to the surface and geometry of the electrode, which makes it difficult to accept that the top surface is better for intercalation than the edges, if indeed other factors come into play.

Our Response:

As we have described in the main text, the reversible intercalation in sealed-edge MoS₂ can be attributed to two reasons: (1) *the flake was clamped and stabilized by the surrounding electrodes preventing structural deformation at the edges of the flake*; and (2) *the diffusion pathways on the top surface are uniformly distributed and mechanically inextensible, which naturally control the intercalation homogeneity compared to the intercalation through the edges where all ions flooded into the opening van der Waals gaps*.

Before explaining the better cycling stability of sealed-edge MoS₂, we need to understand the lower stability of open-edge MoS₂ first. When the Li ions are electrochemically intercalated, the lattice would be expanded in the c-axis direction due to the weak van der Waals bonding. Since the ion intercalation starts from the edge to the center (as shown in Fig. 2b of the main text), the lattice mismatch between the edge and center would likely cause more serious expansion, even exfoliation on the edge. In this case, the lattice could never be restored when Li ions are deintercalated, leading to decayed reversibility. However, when all the MoS₂ edges are sealed, the Li ion can only be intercalated through the top diffusion pathways, which are vacancy defects between covalent bonds. They are uniformly distributed and mechanically stronger than the van der Waals bonding. In addition, the intercalation process is layer by layer from top to the bottom, resulting in more uniform lattice expansion in c-axis direction. Moreover, the surrounding Au electrodes not only block the Li intercalation through the edges, but also clamp the edges and prevent serious expansion due to the superior malleability of Au.

We hope the above explanations have relieved the difficulty to accept that the top surface is better for intercalation than the edges.

Comment: Overall, it is an interesting idea and a bold assertion. However, the experimental data just does not justify the claims. It may indeed be valid and correct but as presented there is just not "there there". Major characterization efforts are necessary and lot more cycling and comparisons would be warranted.

Our Response: We thank the reviewer for highly positive comment on our idea and many suggestions to improve our paper. In the new manuscript, we have carefully discussed the defect densities and added more data for cycling stability. We believe that our manuscript is now greatly improved.

REVIEWER #2

Comment: The authors successfully demonstrated reversible ion intercalation through top surface of few-layer MoS₂ with all edges sealed and confirmed that the top-surface intercalation is more reversible and stable compared to edge intercalation. The authors further investigated the optical and transport properties of the intercalated MoS₂. These results are helpful for developing optoelectronic and nanoelectronics devices and deserve publication.

Our Response: We thank the reviewer for his/her positive comments. The questions and suggestions raised by Reviewer 2 are extremely important and helpful for improving the quality of our work. As will be shown below, we took to heart each recommendation and carried out a series of new measurements. The main conclusion of our paper is further strengthened, and we believe the quality of the paper is significantly improved.

Comment: The uniformity of top-surface intercalated sample in thickness direction. The top-surface intercalation requires more energy, could it be able to uniformly intercalate the sample? How long it may take to fully intercalate the sample? Such information is important, the author should confirm the uniformity of the sample. For example, please provide the surface sensitive optical properties of the bottom surface (making device on a cover glass).

Our Response: We thank Reviewer 2 for raising this question. Following the reviewer's suggestion, we re-examined the uniformity of intercalation by performing a series of Raman measurements on sealed-edge MoS₂ flakes (1) on SiO₂/Si substrate with Si Raman peak as reference and (2) on quartz substrate from backside accessing the bottom surface.

As shown in Figure R6a, the representative Raman peaks of Si substrate were explicitly observed for all the intercalation voltages, which indicated that the 532nm excitation light has totally penetrated the MoS₂ flake and reached Si substrate. For the intercalated state (0.9 V), the E_{2g}¹ and A_{1g} peaks of pristine MoS₂ were totally undetectable, while the intensity of Si peak remained unchanged. This observation further confirmed that the whole MoS₂ flake was completely intercalated and changed to 1T phase. Thus, we can confirm that the MoS₂ sample was uniformly intercalated through top-surface channels without any residual portion of non-affected 2H phase. Furthermore, as suggested by Reviewer 2, we fabricated the MoS₂ device on the transparent quartz substrate and carried out the Raman measurements by illuminating the excitation laser light through the bottom of the substrate directly on the sealed-edge MoS₂ flake. As shown in Figure R6b, when the sample was intercalated at V = 0.9 V, no intrinsic E_{2g}¹ and A_{1g} peaks of pristine MoS₂ could be detected, indicating the uniformity of intercalated MoS₂ flake once again.

For the total intercalation time through top surface, the estimated average time is around 20 seconds from the empirical observations. When we rapidly changed the voltage from 1.0 V to 0.9 V, then acquired Raman signals within 5 seconds and obtained the intercalated spectra with both 1T- and 2H-phase characteristic peaks. However, after the next two or three acquisitions, the 2H-phase peaks (E_{2g}¹ and A_{1g}) would disappear gradually. Therefore, we believe the intercalation time is on the order of 20 seconds (would also depend on the film thickness), which is an empirical estimation.

Figure R6 Raman spectra of sealed-edge MoS₂ flakes. (a) On SiO₂/Si substrate, the excitation laser light was incident on the top surface of the flake with Si Raman peak as a reference. (b) On quartz substrate, the excitation laser light was incident from the bottom of the device. (This figure is now included as Figure S6)

Change to Manuscript: We discussed the uniformity in the main text and add the Figure R6 in supplementary information as Figure S6.

Page 8-9:

“Finally, to examine the uniformity of the intercalation from the top surface, we performed a series of Raman measurements on sealed-edge MoS₂ flakes (1) on SiO₂/Si substrate with the Si Raman peak as reference and (2) on quartz substrate from the backside accessing the bottom surface (**Fig. S6**). We found that the representative Raman peaks of Si substrate were explicitly observed for all the intercalation voltages, which indicates that the excitation laser light has totally penetrated the MoS₂ flake and reached Si substrate (**Fig. S6a**). For the intercalated state, the E_{2g}^1 and A_{1g} peaks of pristine MoS₂ were completely undetectable, while the intensity of Si peak remained unchanged. Thus, we can confirm that the sealed-edge MoS₂ flake was uniformly intercalated through the top surface. We also fabricated the sealed-edge MoS₂ device on a transparent quartz substrate and carried out the Raman measurements by illuminating the excitation laser light through the bottom of the substrate directly to the sealed-edge MoS₂ flake. When the sample was intercalated, no intrinsic E_{2g}^1 and A_{1g} peaks of pristine MoS₂ could be detected, further confirming the uniformity of the intercalation of MoS₂ flake from top surface (**Fig. S6b**).”

Comment: Because the top-surface intercalation is based on the existence of natural defects in exfoliated MoS₂ flake, is it applicable to other 2D materials?

Our Response: We thank Reviewer 2 for this important question. In addition to MoS₂, we have used MoSe₂ as an example to show that our conclusion can be generalized to similar material systems (Figure R7). In the experiments, we observed intercalation from top surface of MoSe₂ with uniform color change and reversible Raman spectra change.

Figure R7 Intercalation of sealed-edge MoSe₂. (a) *In-situ* optical microscopy images of Li intercalation into MoSe₂ through top surface. (b) *In-situ* Raman spectra of Li intercalation into MoSe₂ with sealed edge. (This figure is now included as Figure S23)

Change to Manuscript: We have added discussion on other material systems using MoSe₂ as an example.

Page 15:

“We note intercalation through top surface also applies to other similar 2D layered material systems such as MoSe₂ (Fig. S23).”

Comment: The ion intercalation rate (charging rate) also plays an important role in reversibility and uniformity of the device, please discuss for both top-surface and edge intercalation.

Our Response: We thank the reviewer 2 for raising this comment.

We agree that the intercalation rate is an important factor in determining the reversibility and uniformity of the device. Generally, the higher intercalation rate would produce lower reversibility. In our case, it is impossible to use the constant-current method to control the intercalation rate on a small MoS₂ flake, because a tiny amount of charge (1.52×10^{-9} C) is enough to charge a MoS₂ flake ($10 \mu\text{m} \times 10 \mu\text{m} \times 5 \text{nm}$) to Li₁MoS₂, only a few seconds (1.52 s) are needed if the current is 1 nA. Therefore, we have to use the cyclic voltammetry to control the intercalation procedures. When the intercalation voltage is achieved, the intercalation rate is mainly determined by the device geometry. From our observations, the intercalation rate of open-edge samples is about twice as high as the sealed-edge ones due to the much lower intercalation barrier through the Van der Waals gaps.

Change to Manuscript: We have added the factor of intercalation rate to the discussions of the reversible intercalation in sealed-edge MoS₂.

“(3) the relatively low intercalation rate of sealed-edge MoS₂ may cause less lattice distortion and expansion.”

Comment: Because this work focuses on top-surface intercalation, to highlight its advantage, please provide the transport properties of the top-surface intercalated devices, instead of the edge-intercalated devices. Comparing their difference is also helpful. This is helpful but not required.

Our Response: We thank reviewer 2 for the helpful suggestion. To study the transport properties of the top-surface intercalated devices, we need to seal the MoS₂ flakes using dielectric materials instead of Au. We have fabricated the devices with insulating SiO₂ (100 nm thick) to seal the edges of MoS₂ as shown in the inset of Figure R8a. Compared with the transport data of open-edge MoS₂ (Figure R8b, the same device as in Fig. 2b in the main text), the drain-source current both showed similar dependence on the potentials vs Li/Li⁺. When the potential was first scanned from 3.5 V to 1.5 V, the increase of current results from Li-ion gating effect. From 1.5 V to 1.0 V, the Li ions started to intercalate, and current dropped possibly due to the creation of more defects from Li-ion intercalation. Between 1.0 V and 0.8 V, the current showed a sharp peak, which is probably due to the phase transition from semiconducting 2H phase to metallic 1T phase, giving rise to the drastic current increase. The degradation of Li-intercalated MoS₂, such as exfoliation or other irreversible reactions, may result in the prominent decrease of current.

Due to the similarity of the current curves between sealed- and open-edge devices, we can conclude that the SiO₂ sealant didn't play an essential role during Li-ion intercalation. This is because the SiO₂ layer may be cracked or peeled off during the Li intercalation due to the c-axis expansion and low malleability of SiO₂ compared with Au metal. From the captured images (lower panel of Figure R8a), we found that the original morphology of the SiO₂-sealed MoS₂ device cannot be restored after the retraction of voltage to 3.0 V, which further demonstrated the instability of SiO₂ sealant during Li-ion intercalation. We also tried to use LiF to seal MoS₂, but cannot obtain any positive results at this stage. Thus, we realized that it is extremely difficult to find an insulating material with high malleability, electrochemical stability and high affinity to seal MoS₂ edges.

Suppose that we made such devices, the drain-source current would possibly show a sharp increase and then reach a maximum value rather than a peak in the open-edge devices from 1.0 V to 0.8 V, similar to the Na⁺ intercalated case in Fig.5d in the main text.

Figure R8 Transport property comparison of Li intercalation in sealed and open-edge MoS₂. (a) Source-drain current scan of Li intercalation into MoS₂ sealed with SiO₂. (b) Source-drain current scan of Li intercalation into open-edge MoS₂ (This figure is now included as Figure S20)

Change to Manuscript: We add Fig. R8 and these discussions into the SI. We also added a sentence in the main text.

Page 14:

“We note that it is challenging to find an insulating material with high malleability, electrochemical stability and high affinity to seal MoS₂ edges for the electrical measurements at this point (Fig. S20).”

Comment: Please provide the Hall data (Hall voltage vs Magnetic field at respective potentials and temperatures) in the supplementary information. The potential was applied at 200K, why the carrier density changes as a function of temperature at fixed potential (the electrolyte was frozen at low temperature)?

Our Response: We thank the reviewer for pointing out this issue. The following figure shows the Hall resistance as a function of the magnetic field.

Figure R9 The Hall resistance as a function of the magnetic field at different temperatures and potentials vs NaCoO₂ counter electrode. (*This figure is now included as Figure S22*)

Actually, the potential was applied at 300 K when the electrolyte is liquid, same as that for the Raman and drain-source current measurements in Fig. 2, 3, and 5d in the main text. We apologize for this mistake, and the main text has been corrected now. At fixed potential, the carrier density decreases for lower temperatures because the thermally activated charge carriers are reduced at low temperatures. Below the freezing temperature of electrolyte, the electric-double-layer gating effect as well as the intercalated state would remain unchanged resulting from the immovable ions in frozen electrolyte. To further check the functionality of Na⁺ electrolyte at low temperatures, we performed the 4-probe resistance measurements at different temperatures. As can be seen in Figure R10, the resistance of the MoS₂ flake showed a dramatic decrease when the Na⁺ was intercalated, and then recovered when Na⁺ was de-intercalated. However, higher (more negative) intercalation potentials were needed at lower temperatures, that is -2.7 V at 270 K and -3.2 V at 260 K. At 240 K, the Na⁺ cannot be intercalated even up to -5 V. Therefore, we had to apply the potential at room temperature to

perform the intercalation, and then lowered the temperature and kept the potentials to study the carrier density changes.

Figure R10 The four-probe resistance measurements at different temperatures with respect to the potentials relative to the NaCoO_2 . The arrows indicate the scanning directions. (This figure is now included as Figure S21)

Change to Manuscript: We put the Hall data to the SI and corrected the mistake in the main text.

Figure 5 caption:

“The potential of MoS_2 was applied at 300 K.”

Page 14:

“To eliminate the effect of contact resistance and study the intrinsic transport properties, we fabricated another device with standard Hall-bar geometry and changed the applied potentials only at 300 K, higher than the ineffective temperature of electrolyte (**Fig. S21**). The four-probe resistivity (**Fig. 5e**) reduced exponentially till nearly saturated due to the surface charging from electric-double-layer effect when the potential of MoS_2 was continuously swept from -0.8 V to -2.3 V. As the sample was cooled down, all the resistivity decreased and displayed metallic behavior down to 2 K. When the Na^+ ions were intercalated at the potential of -2.6 V, the resistivity decreased by an order of magnitude compared with that prior to the intercalation at -2.3 V. From the Hall effect measurements (**Fig. S22**), we found that the electron density at -2.3 V can reach $1 \times 10^{14} \text{ cm}^{-2}$ at 2 K (**Figure 5f**), consistent with that in ionic-liquid gated MoS_2 .^{9,45}”

REVIEWER #3

Comment: The authors report on electrochemical ion intercalation as a potential materials synthesis route for tunable optoelectronic, energy and separation properties of 2D materials. They demonstrate this with MoS₂ as an example case where they show reversible intercalation with sealed edges and argue based on a suite of experimental and DFT calculations that smaller cations can "intercalate" while larger cations cannot through the top surface. The manuscript is reasonably well-presented and the findings are certainly intriguing.

Our Response: We thank the reviewer for his/her positive comments. We have carried out a series of new calculations and measurements following Reviewer 3's suggestions and comments. We believe the quality of our manuscript is now significantly improved.

Comment: The use of "plain vanilla" nudged elastic band type calculations to discuss barriers in the context of an electrochemical intercalation reaction is deeply flawed. This is because the work function of the system needs to be held constant and electrochemical reaction barriers simply cannot be estimated in the way done in Figs S6-S10 as the trajectory taken by NEB does not guarantee constant electrochemical potential. Please see for e.g., Chemical Physics Letters, 466, 68-71, J. Phys. Chem. C, 112 (2008), pp. 8747-8750, Phys. Rev. B, 73 (2006), p. 115407, etc. While they may have some data value, they provide no direct comparison to the experimental measurements carried out in this work.

Our Response: We thank Reviewer 3 for this comment. The comment reminds us of a new way to establish a model to simulate the charged surface and calculate the free energy of electrochemical reaction.

In the papers mentioned by the reviewer, several methods have been developed to describe the electrochemical reaction with charge transfer on the surface. Rossmeisl *et al.* took the Heyrovsky reaction as an example to develop a method for calculating the reaction energy of charge transfer reaction and avoid the finite-size effect (Chem. Phys. Let., 2008, 466, 68-71). Jinnouchi *et al.* used the modified Poisson-Boltzmann (MPB) theory as implicit solution and a few explicit water molecules to investigate the factors, such as entropy and charged surface, which can affect the redox potential for surface reaction (J. Phys. Chem. C, 2008, 112, 8747-8750). Otani *et al.* developed a new method for charged surface that combines Poisson equation and Green function with Kohn-Sham equation to remove the restriction of the periodic boundary condition (PBC) of DFT-PW-PP scheme (Phys. Rev. B, 2006, 73, 115407). All these methods are extremely useful for the electron transfer reaction in charged surface. However, there are some differences between the electrochemical reaction investigated in these papers and the intercalation we considered in this manuscript.

In our experiment, the applied potentials drive alkali ions to accumulate on the top of MoS₂ surface before the intercalation. Then the change in potential will provide a driving force to induce the intercalation of alkali ions. It is worth noting that the potential in experiment is carefully controlled in the processes of both intercalation and de-intercalation, insuring that the alkali ions would not be reduced to alkali metal. Hence, the intercalation of alkali ions here is very different from the electrochemical reaction with charge transfer such as hydrogen evolution reaction. In DFT calculations, the charge transfer from Li, Na and K atoms to MoS₂ surface, V_S , V_{S_2} and V_{M_0} at the favorable adsorption sites are all around 0.9 e by Bader analysis, indicating that the alkali atom on MoS₂ surface, V_S , V_{S_2} and V_{M_0} can be regarded as alkali ions. Because of no conversion of Li⁺/Li, we can simplify the intercalation and de-intercalation of alkali ions to the migration of alkali atom from the top of MoS₂ to the bottom. The nudged elastic band (NEB) method we employed in our manuscript has been widely used to explore the migration path in the process of interaction and electrochemical reaction (Phys. Chem.

Chem. Phys., 2007, 9, 3241-3250; Energy & Environ. Sci., 2011, 4, 3680; Nat. Commun., 2015, 6, 6929). Generally, for bulk materials, the electrochemical potential is calculated by the energy change before and after the electrochemical reaction; but for layered materials, it is more suitable to apply the NEB method to study the intercalation behavior of ions through these materials. By comparing the barriers obtained by NEB calculation, we can determine the vital defect for the selective intercalation of alkali ions. This is the main purpose for our DFT calculation.

As suggested by the reviewer, the charged surface may play an important role in the migrating behavior of alkali ions. In response to the comment, we also try to simulate a charged surface by placing a layer of full coverage of Li^+ on MoS_2 surface, which could change the work function of substrate. Without Li^+ coverage, the work function of MoS_2 is 5.63 eV. While the work function will change to 4.60 eV with additional Li^+ coverage. Similar to the additional H^+ coverage to simulate the charged surface for HER/HOR, the Li^+ coverage can also provide a charged surface for the intercalation of Li^+ through MoS_2 . Due to the addition of lithium layer, the inversion symmetry of monolayer MoS_2 is broke down. Here, we decided to compare the migration process of intercalation with the largest barrier on charged surface with that on uncharged surface. Then we calculated the migration barrier of Li^+ in the condition of charged surface. As shown in **Figure R11**, the barrier is 1.24 eV, just 0.06 eV smaller than the neutral surface (barrier as 1.30 eV). We believe the charged surface has limited effects on the intercalation barrier of alkali ions. Therefore, the barrier we obtained in neutral MoS_2 surface can be used to deduce the vital defect for selective intercalation of alkali ions.

As we discuss above, the intercalation of alkali ions can be achieved by overcoming their migration barrier from top of MoS_2 to bottom, which is related to the potential change corresponding to the intercalation voltage vs. open circuit voltage (OCV). From DFT calculations, the most favorable and reasonable vacancy for the selective intercalation of Li and Na ions is V_{Mo} . The energy barrier of intercalation for Li, Na and K ions are 1.30 eV, 0.79 eV and 2.46 eV, respectively. The significant increase of intercalation barrier for K ion predicts the less probable intercalation of K ion and thus the selectivity of the intercalation. In intercalation experiments, the intercalation voltage vs. OCV are ~ 2.2 V (0.8 V vs. 3.0 V) and ~ 2.4 V (-2.4 V vs. 0 V) for Li and Na ions, respectively, and unavailable for K ion till ~ 3.0 V (-3.0 V vs. 0 V). Although the experimental values were not exactly as in the calculations but were in reasonable match. Overall, we believe the DFT calculations have successfully explained the experimental phenomena: namely, how intercalation happens and why intercalation is selective.

Figure R11 Intercalation process of Li^+ on the neutral and charged surface with largest barrier. (This figure is now included as Figure S14)

Change to Manuscript: We will add the detailed analysis of intercalation process in the main text and put the discussion of the effects of charged surface into SI.

Page 10-11:

“To uncover the underlying mechanism, we propose that Li^+ and Na^+ intercalate through the natural defects^{36,37} into sealed-edge MoS_2 . We test our hypothesis by comparing the energy barriers for alkali ions to penetrate a monolayer MoS_2 with and without defects using density function theory (DFT) calculations (Fig. 4a). In experiment, an initial applied potential drives alkali ions to accumulate on the top of MoS_2 surface, and these alkali ions maintain their ionic state during the intercalation and de-intercalation. In addition, due to the inversion symmetry of monolayer MoS_2 , the kinetics dominate over the thermodynamics in the process of intercalation. Therefore, the nudged elastic band method that is widely used to study the kinetic effect in electrochemical reaction³⁸ (*Phys. Chem. Chem. Phys.*, 2007, 9, 3241-3250) is employed here.”

Comment: The authors need to microscopy write down how the process is occurring: where is the electron transfer occurring?

Our Response: We thank reviewer 3 for this comment. How the process occurs is important for us to understand the intercalation and de-intercalation of alkali ions.

In DFT calculation, the intercalation reaction starts from energy favorable adsorption of alkali atom on the MoS_2 surface or vacancy. After the adsorption, the electron transferring from alkali atoms to MoS_2 is around 0.9 e, so these adsorbed alkali atoms can be regarded as alkali ions. Then, we use the migration of adsorbed alkali atom from top of MoS_2 to bottom to simulate the intercalation. Here, we only take monolayer MoS_2 into consideration to investigate the intercalation process of alkali ions. Due to the inversion symmetry of monolayer MoS_2 , the de-intercalation process can be regarded as inverse of the migration of adsorbed alkali atom from top of MoS_2 to bottom. In experiment, there is an initial potential to promote the alkali ions to locate on energy favorable sites, then the potential change will force the alkali ions to intercalate through the layer of MoS_2 by appropriate pathways.

The electron transfer happens in the process of adsorption of alkali atoms, but there is no conversion between Li ion and Li atom in the process of intercalation and de-intercalation.

Change to Manuscript: We added these detailed explanation of intercalation process into SI (Notes of Figure S13).

Comment: How about the thermodynamics of electrochemical ion insertion for these different processes? How does that compare to the experimental numbers? Is it a thermodynamic vs kinetic effect? The authors should estimate this through the standard computational lithium and sodium electrode scheme for concerted ion-coupled electron transfer processes.

Our Response: We thank Reviewer 3 for this comment. When we consider a single layer MoS_2 with and without V_{S_2} and V_{Mo} , due to the mirror symmetry with respect to the basal plane, there is no adsorption energy difference between the initial state and final state in the process of intercalation. That is the reason why most of the calculated NEB energy curves are symmetric. For the layered MoS_2 film, there is van der Waals interaction between different layers, so the energy difference between

the initial and final states is also very small. In order to show the thermodynamics of electrochemical ion insertion directly, we plot the energy profile of the intercalation process of alkali ions through perfect single-layer MoS_2 or MoS_2 with different defects. As shown in **Figure R12**, only V_S (in gray line) breaks the mirror symmetry of the MoS_2 surface, but the energy difference between initial state and final state is negligible compared to its migration barrier. Therefore, the kinetic effect plays vital role in the process of intercalation rather than the thermodynamic effect.

In experiment, an initial potential is applied to promote the adsorption of alkali ions. But the adsorption free energy of alkali ion calculated by DFT is negative, indicating an exothermal adsorption process. This adsorption free energy is incommensurable to the experimental initial potential and can only reflect the stability of adsorbed alkali ions on substrate.

Due to the almost similar adsorption free energy of initial state and final state, the thermodynamic effect is negligible and the kinetic effect is critical in the process of intercalation.

The standard computational method for lithium and sodium electrode is suitable for calculating the intercalation potential in bulk material. It changes the number of the intercalated ions to control the stoichiometric ratio of intercalation compound and get the voltage profile of the intercalation process. However, the monolayer MoS_2 , a 2D layered material, is adopted in our DFT calculation to find the vital defect for the selective intercalation of alkali ions. It is more appropriate to use the NEB method to simulate the intercalation of ions from top of 2D material to the bottom rather than the standard method to calculate the bulk voltage profile.

Figure R12 Energy profile of the intercalation process of alkali ions through perfect MoS_2 , V_S , V_{S2} and V_{Mo} (This figure is now included as Figure S15)

Change to Manuscript: We put the energy plot (**Figure R12**) to SI as Figure S15, and added the analysis of thermodynamics vs. kinetics in the main text and SI (notes of Fig. S15).

Comment: "These diffusion energy barriers through V_{Mo} are on the same order of magnitude as the applied energy in the experiments (difference between OCV and applied voltage)." I am not sure I am convinced of this statement at all, in light of comments above. Also, authors should clarify make the distinction between thermodynamics and kinetics in this case.

Our Response: We thank reviewer 3 for this comment.

As we discuss above, the thermodynamics effect is negligible due to the mirror symmetry of MoS_2 single layer and the kinetics is critical in the process of intercalation. The experimental potential change as the driving force for intercalation should be comparable to the kinetic barrier calculated by DFT. The purpose of our DFT calculation is to determine the vital defect for the selective intercalation of Li, Na and K ions. We take the most typical point defects in MoS_2 into consideration and calculate the adsorption free energy of alkali ions and the kinetic barrier of the intercalation. The thermodynamics is negligible in the process of intercalation, so we summarize the migration barriers (in **Table R1**) and experimental potential change during the intercalation and de-intercalation for comparison.

The experimental potential change of Li, Na and K ions (the intercalation voltage vs. OCV) are ~ 2.2 V, ~ 2.4 V and 3.0 V, respectively. However, only Li and Na ions can achieve the intercalation. From our calculated results, the kinetic barrier of Li, Na and K ion through perfect MoS_2 , V_S and V_{S_2} are much larger than the experimental results. For example, the migration barrier of Li ion intercalating through perfect MoS_2 , V_S and V_{S_2} are 4.03 eV, 4.82 eV and 4.51 eV, which is larger than the experimental potential change of 2.4 V, indicating the less probable intercalation of alkali ions in perfect MoS_2 , V_S and V_{S_2} . Only with the existence of Mo vacancy V_{Mo} , the migration barrier of Li and Na ion through MoS_2 is comparable to the potential change (1.30 eV vs. 2.2 V and 0.79 eV vs. 2.4 V). The migration barrier of K ion through V_{Mo} is much larger than that of Li and Na ion (2.46 eV vs. 1.30 eV & 0.79 eV), indicating the impossible intercalation of K ion. Therefore, we can clarify that V_{Mo} is the vital defect for the selective intercalation of alkali ions.

Table R1 Migration barrier of alkali ions intercalating through perfect MoS_2 , V_S , V_{S_2} and V_{Mo} (in eV) and potential change in experiments (in V) (This table is for reviewers only).

	Perfect (eV)	V_S (eV)	V_{S_2} (eV)	V_{Mo} (eV)	Exp. (V)
Li	4.03	4.82	4.51	1.30	2.2 (intercalation)
Na	8.32	8.94	7.53	0.79	2.4 (intercalation)
K	13.22	13.75	9.14	2.46	3.0 (no intercalation)

Change to Manuscript: We revised the main text to make the statement clear. We added the comparison between experimental and theoretical values.

Page 11:

“These energy barriers for Li^+ and Na^+ to penetrate through MoS_2 is comparable to the potential change (difference between OCV and intercalation voltage) in our experiments (1.30 eV vs. ~ 2.2 V and 0.79 eV vs. ~ 2.4 V). In contrast, the energy barrier for K^+ to go through V_{Mo} is much larger (2.46 eV) compared to Li^+ and Na^+ , explaining the unsuccessful intercalation of K^+ even with ~ 3.0 V potential change in experiment.”

Comment: How many cycles was it possible to reversibly demonstrate the insertion/de-insertion?

Our Response:

We thank Reviewer 3 for pointing out this question. In our experiment, a single nanometer-thick MoS₂ flake is directly exposed to the electrolyte and connected to the Au electrode. It is very different from the cathode materials in Lithium-ion batteries, which are made of composite slurry coated on metal foils (for example, in Wan, J, et al, *Advanced Energy Materials* 2015, 5, 1401742) and can be cycled more than hundreds of times without significant decay of the capacity.

However, the most unique and significant advantage of nano-scale flakes is to *in situ* interrogate the physical and chemical properties during the intercalation process. To the best of our knowledge, our sealed MoS₂ flakes are much more stable than the previous studies on open-edge MoS₂ flakes (Xiong, F, et al. *Nano Letters* 2015, 6777-6784) and on graphite flakes (Bao, W., et al. *Nat. Commun.* 2014, 5, 4224).

To further demonstrate the stability of sealed MoS₂, we carried out the Raman measurements up to 20 cycles. As shown in the following figure, both the E_{2g}¹ and A_{1g} peaks can be clearly identified for all the 20 cycles, indicating the crystalline stability of MoS₂ after Li intercalation and deintercalation procedures. Moreover, the 1T phase peaks from 154 to 207 cm⁻¹ always show up for the intercalated state. Again, Our Raman results further confirm the stability of sealed MoS₂ flakes.

Figure R13 Raman spectra of Li intercalation into sealed MoS₂ flakes up to 20 intercalation and deintercalation cycles. The blue (red) curves represent the Li-deintercalated (intercalated) states. (*This figure is now included as Figure S4*)

Change to Manuscript: We added discussion about the stability up to 20 cycles in the main text as follows:

Page 8:

“In addition, we found both the E_{2g}^1 and A_{1g} Raman modes can be clearly identified for up to the 20 cycles of Li intercalation and de-intercalation processes (**Fig. S4**), indicating the crystalline stability of sealed-edge MoS₂.”

REVIEWERS' COMMENTS:

Reviewer #1 (Remarks to the Author):

Extensive answers and added comments. Thanks. No more info. needed.

Reviewer #3 (Remarks to the Author):

I commend the authors for taking a deep look at the reviewer comments and addressing them extensively. Although, there are a few minor things that I am still not happy with, I am overall happy with the authors addressing the reviewer concerns. In light of this, I think the manuscript can be published.

REVIEWERS' COMMENTS:

Reviewer #1 (Remarks to the Author):

Extensive answers and added comments. Thanks. No more info. needed.

We thank Reviewer 1 for the time and effort in reviewing our manuscript and for providing highly positive evaluation of our work.

Reviewer #3 (Remarks to the Author):

I commend the authors for taking a deep look at the reviewer comments and addressing them extensively. Although, there are a few minor things that I am still not happy with, I am overall happy with the authors addressing the reviewer concerns. In light of this, I think the manuscript can be published.

We thank Reviewer 3 for the highly positive comments to our manuscript and responses. We really appreciate his/her agreement to the publication.